# Urbanization and health: The effects of the built environment on chronic disease risk factors among women in Tanzania

**Jessie Pinchoff**[1]*, **Carrie W. Mills**[2], **Deborah Balk**[2]

**1** Department of Poverty Gender and Youth, Population Council, New York, NY, United States of America,
**2** CUNY Institute for Demographic Research, City University of New York, New York, NY, United States of America

☯ These authors contributed equally to this work.
* jpinchoff@popcouncil.org

**Data Availability Statement:** The data underlying the results presented in the study are available from the Demographic and Health Program (https://dhsprogram.com/data/available-datasets.

## Abstract

Sub-Saharan Africa is experiencing rapid urban growth. Cities enable greater access to health services and improved water and sanitation infrastructure, leading to some improvements in health. However, urban settings may also be associated with more sedentary, stressful lifestyles and consumption of less nutritious food. C-reactive protein (CRP) is a measure of chronic inflammation predictive of cardiovascular disease, and high body mass index (BMI), a ratio of weight to height, indicates overweight or obesity and is associated with an increased risk of many chronic diseases. To explore the association between urbanicity and these two markers, we overlaid data from the 2010 Tanzania Demographic and Health Survey (DHS) with a satellite-derived measure of built environment. Linear regression models were constructed for the outcomes of BMI and CRP, by 1) administratively defined urban/rural categorization from the DHS, 2) satellite derived built environment, and 3) built environment stratified by urban/rural. A total of 2,212 women were included; 23% had elevated CRP, 21% were overweight or obese. A third (33%) lived in a highly built up area and 29% lived in an area classified as urban. A strong positive association between both CRP and BMI and built environment was detected; log CRP increased 0.43 in the highest built up areas compared to not built up (p<0.05); log BMI increased 0.02 in the most built up areas compared to not built up (p<0.05). However, comparing urban to rural category was only significant in unadjusted models. Models stratified by urban/rural category highlight that the variation in CRP and BMI by built environment is mainly driven by rural areas; within urban areas there is less variation. Our findings highlight the potential negative effects of urbanicity on chronic disease markers, with potentially more change detected for those transitioning from rural to urban lifestyles. Satellite-derived urbanicity measures are reproducible and provide more nuanced understanding of effects of built environment on health.

cfm) and from the European Commission (https://ghsl.jrc.ec.europa.eu/).

**Funding:** Co-author Carrie Mills received funding from the CUNY Institute for Demographic Research as a Demography Fellow. Otherwise, the authors received no specific funding for this work.

**Competing interests:** The authors have declared that no competing interests exist.

## Introduction

Urban environments, as well as the process of urbanization, are believed to have both positive and negative effects on health. While overall there is an urban advantage to health on average, in most low-income countries averages mask true differences in health that may be seen by disaggregating by certain factors [1]. Residents of urban environments often have greater access to health care and social services, although access may differ by city size [2] and may not be distributed equally among all residents. Urban environments tend to offer greater access to education and increased job opportunities, both of which can improve health. However, "urbanicity"—the characteristics of a locality being urban–also often leads to a more sedentary lifestyle, less access to fresh foods ('food deserts') and more access to processed food with a poor nutritional profile, and, especially in poorer developed countries, more crowded living situations with greater chance for unsanitary conditions [3–6]. Additionally, while urban life offers many benefits, some argue that urban spaces may create negative psychosocial factors due to experiences of social fragmentation, overcrowding, and crime prevalent in some cities [7]. A study in Burkina Faso found high rates of major depressive episodes among the urban poor due to chronic health problems and poor standard of living [8].

It has been proposed that low-grade systemic and chronic inflammation associated with modern lifestyles and environments may help serve as a unifying theory to understand chronic disease etiology [9]. In particular, the American Heart Association has established that C-reactive protein (CRP) is the best inflammatory marker to use as a proxy for cardiovascular disease and evidence to date supports use of CRP as an independent predictor of increased coronary heart disease risk. Based on studies of the distributions of CRP samples, an approximately two-fold increase in relative coronary risk between high (>3.0 mg/L) and low (<1.0 mg/L) tertiles of CRP has been reported. While most research is on European or European-American CRP samples, a 2014 review study found that CRP was a strong independent predictor of inflammation and heart disease across different ethnic groups [10]. Additionally, research has identified psychosocial factors commonly prevalent in urban environments that may be associated with inflammatory processes, including chronic stress, low social support, and poor psychological health [11–13].

While there is generally a lack of research on non-communicable diseases from sub-Saharan Africa, prevalence estimates indicate growing rates of overweight and obesity, and thus it is expected that these regions will experience increases in chronic health outcomes including diabetes and cardiovascular disease [14–16]. A recent study assessing overweight and obesity among women in Sub-Saharan African countries found a wide variation of rates among countries, with wealth being the strongest predictor in most countries [17]. Studies have also demonstrated significant increases in overweight and obesity over the past 15 years in this region [18]. There are scant descriptive statistics, particularly of CRP but also of overweight or obesity, in countries such as Tanzania. One very small study in Tanzania found CRP levels were strongly associated with coronary heart disease [19]. However, there are few large-scale and no nationally representative studies assessing elevated levels of CRP and measures of overweight and obesity in sub-Saharan African populations.

Available evidence also indicates an association between body mass index (BMI), a metric of weight for height used to classify weight status, and CRP; a systematic review found consistent association between CRP and both overweight and obesity, such that higher BMI is associated with elevated levels of CRP [20]. However, studies included in this review and most others in the literature do not include any participants from sub-Saharan Africa. One study with a very small sample size was identified from the Sudan, where researchers found a strong association between CRP levels and obesity [21]. CRP and BMI are related and also each relates to chronic disease risk.

Most surveys and datasets in both the developed and developing world define urbanicity as a binary variable, with areas classified as being either urban or rural. Often, categorization is made via administrative records and boundaries and are guided by census data collected by national statistical offices every ten years (or even less frequently). However, urbanicity is too complex and multifaceted to be measured using a binary measure. Given the growing need to understand urbanization and its health effects, researchers are exploring novel approaches and methodologies to characterize what is meant by urbanization and urban life [22]. Some researchers have created and employed the use of multi-component indices that seek to capture several aspects that together define an urban space, including physical, social, and economic components in order to understand the effect of these environments on various health outcomes [3, 23–25]. While use of a multi-component index allows for a broader and more nuanced description of the environment, limitations of this method include the laborious or sometimes infeasible requirement to obtain all of the different measures of the index, the need to update them regularly in order to keep up with the fast pace of urban development and growth, as well as the subjective nature inherent in both choosing which indicators to include as well as in the categorization of specific indicators [3].

Satellite remote sensing generally measures either the light or electromagnetic radiation reflected from objects on the earth or the earth itself. While these direct measures of reflected energy can be used to determine various factors such as whether the surface is soil or vegetation, as well as different phenomena occurring at the surface such as fire [26], a useful measure to understand urbanization is an index determining the built-up presence [27, 28]. Satellite remote sensing offers several advantages over multicomponent indices–it is much less subjective, does not require so many different components, and can be integrated with any existing data source containing location data. Dorelien and colleagues in a recent study (2013) find that by combining satellite data with survey data, an urban continuum rather than a dichotomy can be constructed [22]. Similarly, recent work by the European Commission finds that by combining satellite and census data, a degree of urbanization metric allows for understanding the urban continuum [29]. These new approaches allow for much greater differentiation of urban socioeconomic, health, and demographic differences that may occur along this continuum [30].

In order to address the lack of cardio-metabolic description among sub-Saharan African populations, this analysis seeks to identify determinants of elevated levels of CRP and BMI among women of child-bearing age living in Tanzania. Using two separate measures of urbanization, we seek to determine the association between urbanicity and CRP and BMI in this population. We hypothesized that both the bivariate measure of urbanicity and the satellite-derived measure of the built environment would be associated with elevated levels of CRP and high BMI. We further aim to tease apart the impact of urbanization from those of socioeconomic status on CRP and BMI.

## Methods

### Data

The data for this study come from the Tanzania Demographic and Health Survey 2010. Demographic and Health Surveys (DHS) are cross-sectional, nationally representative surveys conducted in many low-income countries in order to produce estimates of demographic, health, and nutritional behavior. The multistage sampling methodology requires a first sampling frame of non-overlapping area units that cover the entire country, proportional to population. A fixed proportion of households within these units, or clusters, is then chosen by random sampling. Global position system (GPS) coordinates for the cluster centroid–that is, an

approximate center of the sampling unit–are collected using a handheld GPS unit during field-work. In order to protect the confidentiality of survey respondents, the survey clusters are ran-domly dislocated up to 2 and 5 km, in urban and rural areas, respectively (confined to major administrative divisions and coastlines) [17].

Interviews are conducted with all women ages 15 to 49 within these households. Of the 9,194 women with complete interviews, blood samples to assess CRP were collected from a 25% subsample of overall participants. Among those eligible for clinical testing, complete sur-vey and clinical data were available for 2,343 women. We restricted to women not pregnant at the time of the survey. We evaluated how our sample differed from the overall survey sample. Compared with those for whom no clinical values were assessed, participants in our study did not differ statistically by any demographic or socioeconomic characteristics or by any mea-sures of urbanicity. Ethical approval was not required as this is a secondary data analysis using publicly available, de-identified data.

### Urbanicity measurements

The DHS assigns a bivariate measure of urbanicity (urban/rural) to each participant based on an administrative assignment of household location determined by the national statistical office. The sample is stratified by this measure in order for the DHS to be nationally representative of all urban and all rural areas. As discussed earlier, previous work has shown that urban-rural clas-sifications create a false dichotomy and that urbanicity is much more of a continuum [22, 31]. This study therefore also explores a second measure of urbanicity, using a satellite-derived mea-sure of built up environment. The Global Human Settlement Layer (GHSL) is a raster dataset, based largely on classification of Landsat data and corrected by sentinel satellite data, that charac-terizes the built environment of the whole planet at a resolution of 250 meters for four time points ('epochs'–i.e., detected by) 1975, 1990, 2000, and 2014 [32, 33]. Each 250 meter grid cell, or pixel, receives an index score of 0 to 100 (fully built up). GHSL differs from other Landsat-based urban classifications in that it is global, aims to measure built-up rather than impervious surface per se or a residual non-urban classification, and covers multiple time periods from 1975–2014. It differs from other satellite data products (such as MODIS and the night-time lights data, which also measure light not vegetation) in that it is a higher resolution. This study included data from the 2014 epoch of the dataset, because the 2014 data set is informed by the newer sentinel data as well as Landsat, and thus the best representation of the built-environment around the time of survey. The 2014 built environment score for maximum values were catego-rized as none detected (index equal to 0%), low (<20%), low-medium (20-<40%), and built-up (40% or greater). Because the 2014 GHSL data were used rather than the 2000 epoch, it is rele-vant to note the mean change in built-up level from 2000–2014 is less than 2 percentage points, and that about two-thirds of areas experience no change at all. Fig 1 shows the largest city in Tan-zania, Dar es Salaam, with GHSL categories overlaid with the DHS urban/rural classified buffers.

### Health indicators

C-reactive protein (CRP) was originally included as a biomarker in the DHS to control for the effect of active infection on vitamin A level in the 2010 DHS assessment of micronutrients [34]. Blood samples were obtained from respondents by a finger prick and placed on a filter paper card, which was stored in a specially designed container to dry overnight. The dried blood spot samples were shipped to the National Public Health Laboratory for processing. Samples were tested in duplicate using a high-sensitivity commercial ELISA test kit (Bender MedSystems GmbH, Vienna, Austria) [34]. Participants with CRP values greater than 20 mg/L were excluded from this analysis (n = 38). Some clinical guidelines have recommended using a

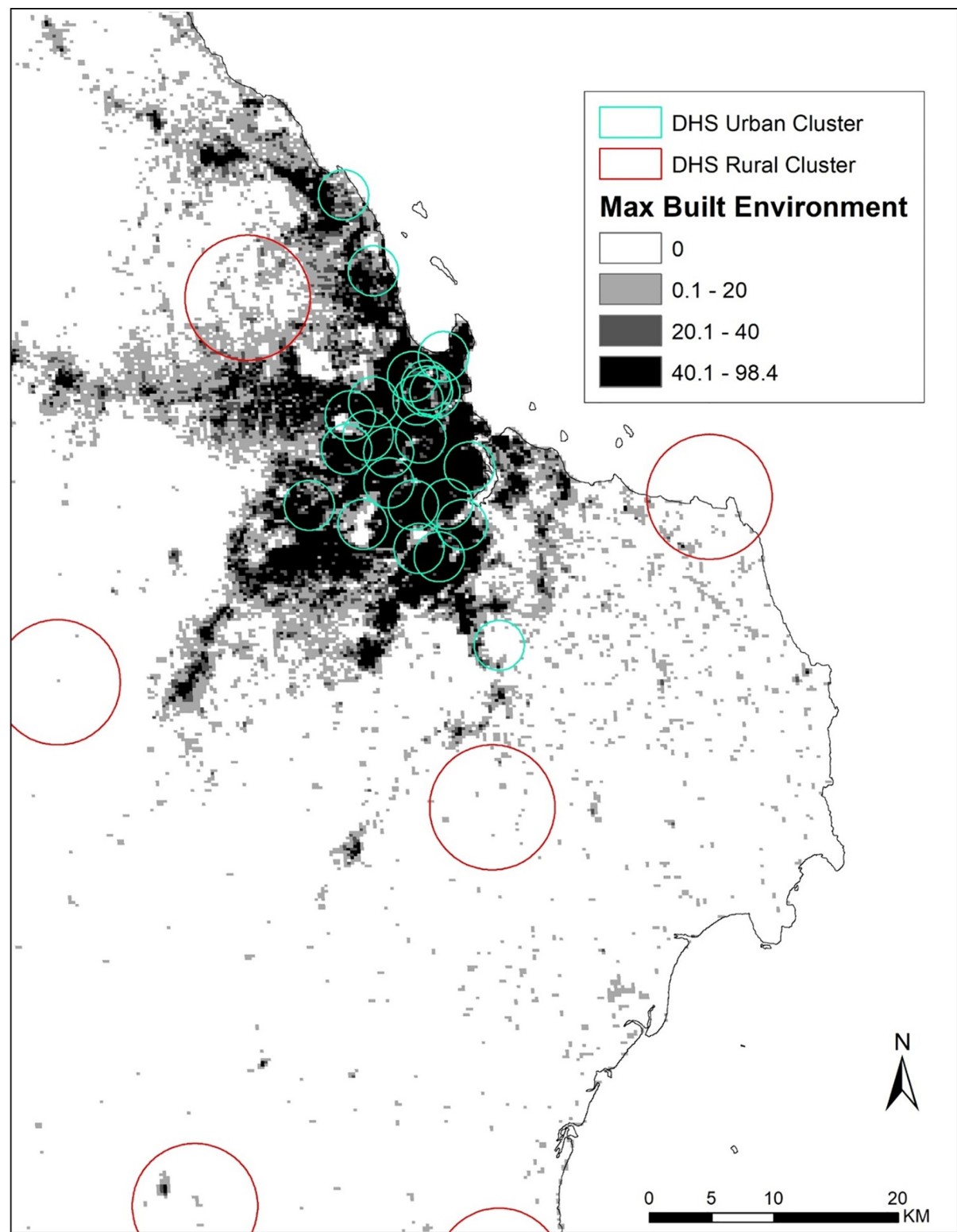

**Fig 1. Spatial data layers, close-up of Dar es Salaam, Tanzania, Demographic and Health Survey 2010 and GHSL maximum built environment from 2014.**

cutoff of 10 mg/L in order to avoid capturing CRP values elevated due to active infection [35]; however, other studies have demonstrated clinical utility in using a higher CRP cutoff [36]. After reviewing the distribution of CRP values among the sample population, we chose to set 20 mg/L as the CRP cutoff. Results using the more conservative cutoff of 10 mg/L were also run but not presented, with similar results.

Each participant underwent anthropometric measurements to obtain height and weight for calculation of BMI (weight in kilograms divided by height in meters squared). Weight categories for descriptive analyses were defined using World Health Organization definitions: underweight was defined as BMI less than 18.5; normal weight, BMI of 18.5 to less than 25; overweight, BMI of 25 to less than 30; and obese, BMI equal to or greater than 30 kg/m$^2$. BMI was modeled as a continuous variable.

### Covariates

Covariates derived from the DHS questionnaire included age, level of education (no school, primary school, or secondary school or higher), marital status (currently married or living together, formerly married, or never married), and wealth index (poorest, poorer, middle, richer, richest). The wealth index is a composite measure that quantifies the economic resources of a household using data regarding selected asset ownership, materials used in housing construction, and types of access to water and sanitation [37, 38].

We also calculated the distance between each centroid cluster and the nearest major city (i.e., area classified as an urban center rather than a town or surrounding semi-dense or exurban area), measured in kilometers from the Degree of Urbanization Data [39, 40].

### Statistical analysis

Descriptive statistics were run describing characteristics of the sample using weighted measures that take into account the two-stage sampling design of the survey and allow for observations to be representative at the national level. In models, CRP was assessed as a continuous measure rather than as a categorical measure, following findings indicating a linear relationship between CRP and risk of cardiovascular disease that did not specifically follow specific threshold effect (e.g., differences in risk at 1.0 or 3.0) [36]. After inspection of the data, CRP was log-transformed for analysis as was BMI.

We tested the association between the bivariate measure of urban/rural and CRP level and BMI using linear regression. Because the environmental variables are constructed at the level of the survey cluster, our regressions used cluster-robust methods to assess intraclass correlation and obtain robust standard error. Bivariate models were run followed by fully adjusted models including all covariates. We then tested the association between the satellite-derived measures of urbanicity and CRP and BMI, also using linear regression and the same approach used for urban/rural. In a final model, we explored the association between GHSL built environment and CRP and BMI stratified by urban or rural classification.

We conducted two sensitivity analyses not presented here. First, all models were run excluding individuals with CRP greater than 10 mg/L. Second, all models were run excluding participants who were underweight. We also ran post-hoc exploratory analyses to test the interaction between wealth and measure of urbanicity. Analysis was conducted using R statistical software and ArcGIS version 10.5.

### Results

A total of 2,212 female participants were included in the overall sample, with a mean age of 29 (range, 15 to 49 years) (Table 1). Half of all women (50.5%) had CRP levels that were low,

**Table 1. Characteristics of participants included in the study.**

| | N | % |
|---|---|---|
| N. | 2212 | |
| Age (mean/SD) | 29.0 | 10.0 |
| CRP Measure | | |
| Low (0-<1.0 mg/L) | 1153 | 50.5% |
| Average (1.0-<3.0 mg/L) | 584 | 26.5% |
| Elevated (3.0-<20.0 mg/L) | 475 | 23.0% |
| Body Mass Index (BMI) Category | | |
| Underweight (BMI <18.5 kg/m$^2$) | 270 | 11.1% |
| Normal weight (18.5 ≤ BMI <25 kg/m$^2$) | 1454 | 68.0% |
| Overweight (25 ≤ BMI < 30 kg/m$^2$) | 325 | 14.6% |
| Obese (BMI ≥ 30 kg/m$^2$) | 157 | 6.2% |
| Wealth Index (Quintile) | | |
| Poorest | 345 | 16.9% |
| Poorer | 422 | 19.6% |
| Middle | 387 | 17.8% |
| Richer | 499 | 20.2% |
| Richest | 559 | 25.5% |
| Education category | | |
| No education | 387 | 17.5% |
| Primary | 1282 | 65.9% |
| Secondary+ | 543 | 16.6% |
| Marital Status | | |
| Never | 665 | 27.2% |
| Currently / living together | 1313 | 61.8% |
| Formerly married | 234 | 11.0% |
| Location from DHS | | |
| Rural | 1649 | 71.4% |
| Urban | 563 | 28.6% |
| Maximum built environment (2014) | | |
| None detected (0%) | 363 | 18.1% |
| Low (1-<20%) | 794 | 38.6% |
| Low-medium (20-<40%) | 301 | 10.4% |
| Built-up (40%+) | 754 | 32.9% |
| Distance to nearest city in kilometers (mean/SD) | 390 | 380 |

Frequencies (N) are unweighted, percentages and means are weighted.

26.5% had average CRP levels, and 23.0% had elevated levels of CRP. The majority of the women (68.0%) were normal weight, while 11.1% were underweight and 20.8% were either overweight or obese. Seventeen percent of women scored in the poorest category of the wealth index while a quarter (25.5%) fell in the richest category. Most (65.9%) had completed primary school and were currently married or living with a partner (61.8%).

Seventy-one percent resided in areas administratively defined as rural, 28.6% urban. The maximum built environment index indicated that 18.1% of women lived in areas with no built environment detected, 38.6% lived in areas classified as low built up, 10.4% lived in low-medium areas, and the remaining 32.9% lived in highly (over 40%) built-up areas. Most urban defined clusters were also in the highest built up category (81.8%), however the rural defined

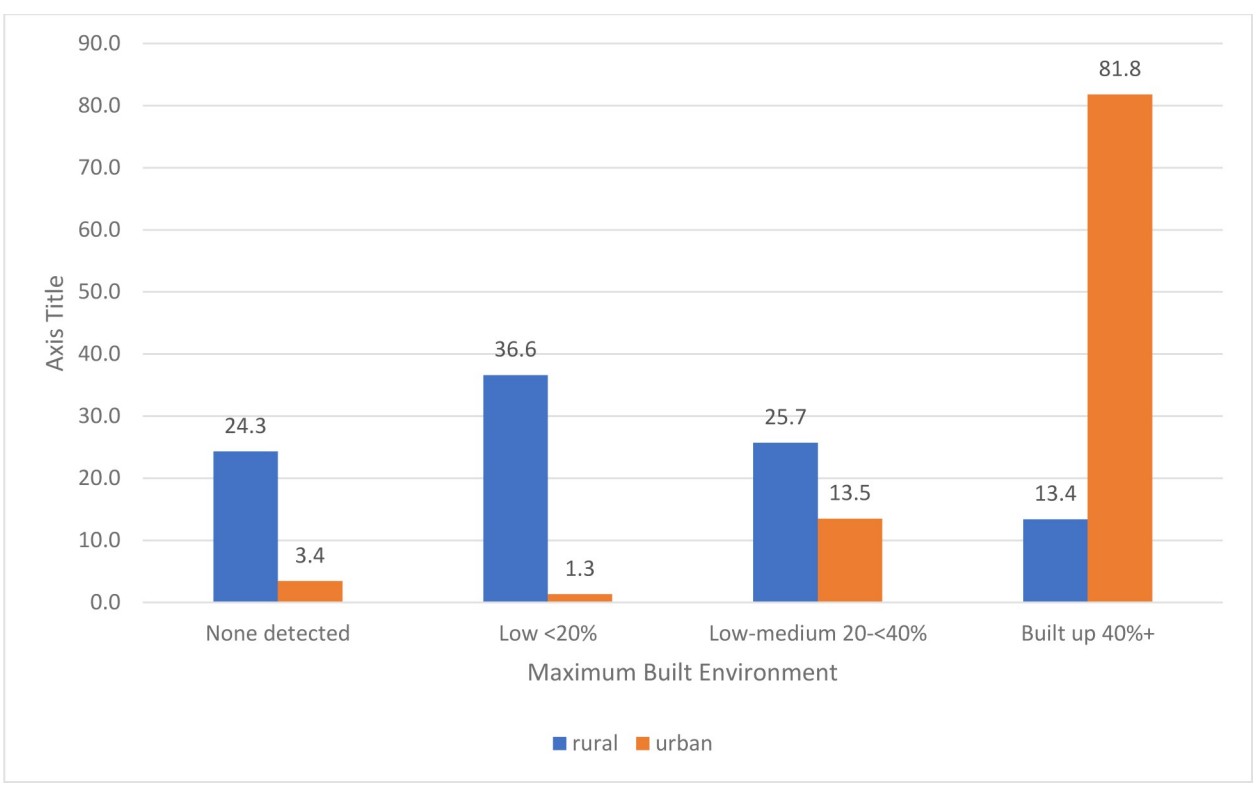

**Fig 2. Distribution of the maximum built environment across urban and rural locations.**

clusters had more variation with over half in either the low-built up category (36.6%) or 0 built up (24.3%) (Fig 2).

Table 2 tabulates and calculates weighted chi square p-values for comparing elevated to non-elevated CRP, and overweight/obese vs normal weight by three different measures of

**Table 2. Prevalence of elevated CRP and overweight/obesity by urban vs. rural location and maximum categories of the built environment (and tests of significance, p-value).**

|  | Elevated CRP | Overweight/Obese |
|---|---|---|
| DHS Classification |  |  |
| Rural | 21.2 | 17.6 |
| Urban | 27.7 | 37.0 |
|  | p = 0.01 | p<0.0001 |
| Built up area classification |  |  |
| None detected | 22.0 | 12.9 |
| Low (1–20%) | 21.9 | 18.5 |
| Low-medium (20-<40%) | 19.0 | 20.4 |
| Built-up (40%+) | 26.3 | 35.1 |
|  | p = 0.17 | p<0.0001 |
| Distance to the nearest city (quartiles) |  |  |
| Quartile 1 (nearest) | 26.0 | 34.7 |
| Quartile 2 | 19.2 | 21.2 |
| Quartile 3 | 24.7 | 19.9 |
| Quartile 4 (farthest) | 21.6 | 15.8 |
|  | P = 0.10 | P<0.0001 |

urbanicity. For the urban/rural classification, 27.7% of participants in urban clusters had elevated CRP compared with 21.2% of participants in rural clusters (p = 0.01); the proportion overweight/obese was also higher among urban participants (37.0%) compared to rural (17.6%) (p<0.0001) (Table 2). By built up area categories, the proportion with elevated CRP in the highest built up category was higher (26.3%) compared with the lowest built up category (22.0%) but this was not statistically significant. Proportion overweight/obese demonstrated a linear increase from 12.9% in the lowest built up category to 35.1% in the highest (p<0.0001). The opposite pattern was found when tabulating quartile distance from the nearest big city; those closest to or in a major city had a higher proportion overweight/obese (34.7%) and with elevated CRP (26.0%) compared with those the farthest from a major city (15.8% and 21.6%, respectively), although this was not significant for CRP. These associations were next tested with linear regression using continuous indicators of CRP and BMI.

The effect of living in an urban location, as measured by the DHS dichotomous variable, was found to increase the log CRP by 0.31 (p < .01) (Table 3). However, in the adjusted model also controlling for age, wealth, education, marital status, and distance to the nearest city, the effect of urban residence was no longer significant. The built environment was associated with CRP in both unadjusted and adjusted models. In the unadjusted model, compared with none detected/0% built up, as category of built environment increased CRP also increased. In the highest built up category compared with 0%, log CRP increased by 0.55 (p<0.01). This was also the case within the adjusted model although the association was not linear. Compared

**Table 3. Unstandardized beta coefficients from linear regression models of the association between measures of urbanicity and log CRP and log BMI.**

| | Log CRP | | | | Log BMI | | | |
|---|---|---|---|---|---|---|---|---|
| | Urban/Rural | | Built environment | | Urban/Rural | | Built environment | |
| | Model 1a | Model 2a | Model 3a | Model 4a | Model 1b | Model 2b | Model3b | Model 4b |
| | b | b | b | b | b | b | b | b |
| Location (urban vs rural) | 0.314** | 0.181 | - | - | 0.083*** | 0.001 | | |
| Built Environment (0%) | - | - | REF | REF | - | - | REF | REF |
| 1–20% | - | - | 0.389* | 0.406* | - | - | 0.027* | 0.019* |
| 20–40% | - | - | 0.504* | 0.480* | - | - | 0.037* | 0.021 |
| >40% | - | - | 0.551** | 0.431* | - | - | 0.093*** | 0.024* |
| Mean distance to city | - | 0.168 | - | 0.169 | - | -0.021* | - | -0.019^ |
| Age (in years) | - | 0.018** | - | 0.019** | - | 0.002*** | - | 0.002*** |
| Education (None) | - | REF | - | REF | - | REF | - | REF |
| Primary | - | -0.056 | - | -0.054 | - | 0.010 | - | 0.010 |
| Secondary or more | - | -0.376* | - | -0.370* | - | 0.005 | | 0.005 |
| Marital status (Never Married) | - | REF | - | REF | - | REF | - | REF |
| Formerly married | - | 0.196 | - | 0.194 | - | 0.028^ | - | 0.027^ |
| Currently married/Living together | - | 0.349* | - | 0.346* | - | 0.056*** | - | 0.056*** |
| Wealth quintile (Richest) | - | REF | - | REF | - | REF | - | REF |
| Richer | - | -0.267^ | - | -0.320* | - | -0.072*** | - | -0.069*** |
| Middle | - | -0.493* | - | -0.535** | - | -0.104*** | - | -0.099*** |
| Poorer | - | -0.561* | - | -0.618** | - | -0.131*** | - | -0.126*** |
| Poorest | - | -0.479* | - | -0.483** | - | -0.148*** | - | -0.141*** |

^p<0.10

* p<0.05

**p<0.01

***p<0.0001

with 0%, those in the highest built up category had log CRP increased by 0.43 (p<0.05). Wealth in both models had an inverse relationship, with those in the poorest wealth quartile having lower CRP measures. Distance to the nearest city was not significantly associated with log CRP in either model.

Living in an urban area, as defined by the DHS dichotomous variable, increased log BMI by 0.08 (p < .0001) in the crude model; but this relationship was not significant in the adjusted model (Table 4). The built environment was significantly associated with increases in BMI in both the crude and adjusted models. After adjusting for age, wealth, education, marital status, and distance to the nearest big city, living in the highest built environment category increased log BMI by 0.02 (p < .05). Wealth, education, and marriage were all significantly associated with increased BMI values as well. In addition, as distance to major cities increased, log BMI decreased by 0.02 in the unadjusted model (p<0.05) and decreased by 0.02 (p<0.1) in the adjusted model.

In the models stratified by urban or rural category, it is clear that much of the variation in CRP and BMI by built up environment is driven by those in the rural category. Among participants in rural clusters, CRP increased by increasing levels of built environment (although not linearly) (Table 4). Among those in the rural category, log CRP in the most built up category was 0.43 higher compared with 0% built up (p<0.10). Built environment was not associated with CRP among urban-only households. Wealth was not associated with CRP in urban or rural models.

For BMI, a similar pattern was observed between built environment in rural areas. Among rural classified clusters, BMI increased across categories of built environment; those in the

**Table 4. Unstandardized beta coefficients from linear regression models of the association between log CRP and built environment stratified by urban/rural classification.**

| | Rural | | Urban | |
|---|---|---|---|---|
| | b | 95% Confidence Interval | b | 95% Confidence Interval |
| Built environment (0%) | REF | REF | REF | REF |
| 1-<20% | 0.439* | 0.062,0.816 | -0.374 | -1.203,0.455 |
| 20-<40% | 0.618** | 0.242,0.995 | -0.634 | -1.526,0.259 |
| 40%+ | 0.431^ | -0.026,0.888 | -0.384 | -1.058,0.291 |
| Distance to nearest city | 0.228 | -0.056,0.511 | 0.075 | -0.512,0.662 |
| Age (in years) | 0.017* | 0.004,0.031 | 0.018 | -0.009,0.045 |
| Education (none) | REF | REF | REF | REF |
| Primary only | -0.019 | -0.299,0.262 | -0.249 | -0.757,0.259 |
| Secondary or above | -0.716** | -1.174,-0.257 | -0.223 | -0.775,0.330 |
| Never Married | REF | REF | REF | REF |
| Formerly married | 0.186 | -0.280,0.652 | 0.091 | -0.917,1.099 |
| Currently married/Living together | 0.281^ | -0.051,0.612 | 0.484^ | -0.021,0.988 |
| Wealth quintile (richest) | REF | REF | REF | REF |
| Richer | 0.052 | -0.410,0.513 | -0.483* | -0.943,-0.022 |
| Middle | -0.285 | -0.773,0.203 | -0.48 | -1.171,0.211 |
| Poorer | -0.342 | -0.815,0.131 | -2.552 | -5.615,0.511 |
| Poorest | -0.269 | -0.732,0.194 | 0.122 | -0.687,0.930 |

^p<0.10

* p<0.05

**p<0.01

***p<0.0001

**Table 5. Unstandardized beta coefficients from linear regression models of the association between log BMI and built environment stratified by urban/rural classification.**

| | Rural | | Urban | |
|---|---|---|---|---|
| | b | 95% Confidence Interval | b | 95% confidence interval |
| **Built environment (0%)** | REF | REF | REF | REF |
| 1-<20% | 0.019^ | -0.001,0.038 | 0.053 | -0.045,0.150 |
| 20-<40% | 0.022 | -0.008,0.053 | 0.022 | -0.083,0.126 |
| 40%+ | 0.039* | 0.015,0.063 | 0.007 | -0.082,0.096 |
| Distance to nearest city | -0.025* | -0.048,-0.002 | -0.002 | -0.051,0.047 |
| Age (in years) | 0.001* | 0.000,0.002 | 0.006*** | 0.003,0.009 |
| Education (none) | REF | REF | REF | REF |
| Primary only | 0.011 | -0.010,0.032 | 0.002 | -0.078,0.081 |
| Secondary or above | 0.019 | -0.019,0.056 | -0.006 | -0.089,0.076 |
| Never Married | REF | REF | REF | REF |
| Formerly married | 0.022 | -0.017,0.061 | 0.04 | -0.023,0.103 |
| Currently married/Living together | 0.064*** | 0.039,0.088 | 0.026 | -0.018,0.070 |
| Wealth quintile (richest) | REF | REF | REF | REF |
| Richer | -0.09*** | -0.126,-0.053 | -0.064** | -0.106,-0.023 |
| Middle | -0.116*** | -0.153,-0.078 | -0.061 | -0.146,0.025 |
| Poorer | -0.139*** | -0.176,-0.101 | -0.152** | -0.250,-0.053 |
| Poorest | -0.151*** | -0.188,-0.115 | -0.187*** | -0.247,-0.126 |

^p<0.10

* p<0.05

**p<0.01

***p<0.0001

40% or more built up category had log BMI increased by 0.04 kg/m$^2$ compared to 0% built up (p<0.05) (Table 5). Among rural classified clusters, log BMI decreased by 0.03 kg/m$^2$ for each kilometer increase in distance from a major city (p<0.05). Wealth was highly associated with BMI in both urban and rural defined clusters. In rural areas, the poorest had a 0.15 kg/m$^2$ decrease in log BMI compared to the richest; in urban areas, the poorest had a 0.18 kg/m$^2$ decrease in log BMI compared to the richest.

Because urbanization is associated with wealth, and the DHS wealth measure is associated with urban residence [31], as it includes in its measure commodities and services more likely to be found in urban areas (e.g., particular roof and flooring types, electricity, and municipal water and sewer)–we explored interactions between wealth and urban/rural but these were not statistically significant. Fig 3 highlights the association between quintiles of wealth and proportion in each category with elevated CRP or overweight/obese status; while both outcomes decrease as wealth decreases, the association with wealth is only significant for those who are overweight/obese (Fig 3). In a supplement we include a figure of a box plot of the maximum built up environment measure (continuous) by wealth category stratified by urban/rural location to highlight that rural clusters have much lower wealth overall, but that within rural and urban defined clusters the built up measures decrease from the richest to the poorest quintile, though there are more outliers in rural areas (S1 Fig).

## Discussion

This study suggests that key biomarker and anthropometric indicators are associated with urban lifestyles in Tanzania, comparing results across models that define urban based on

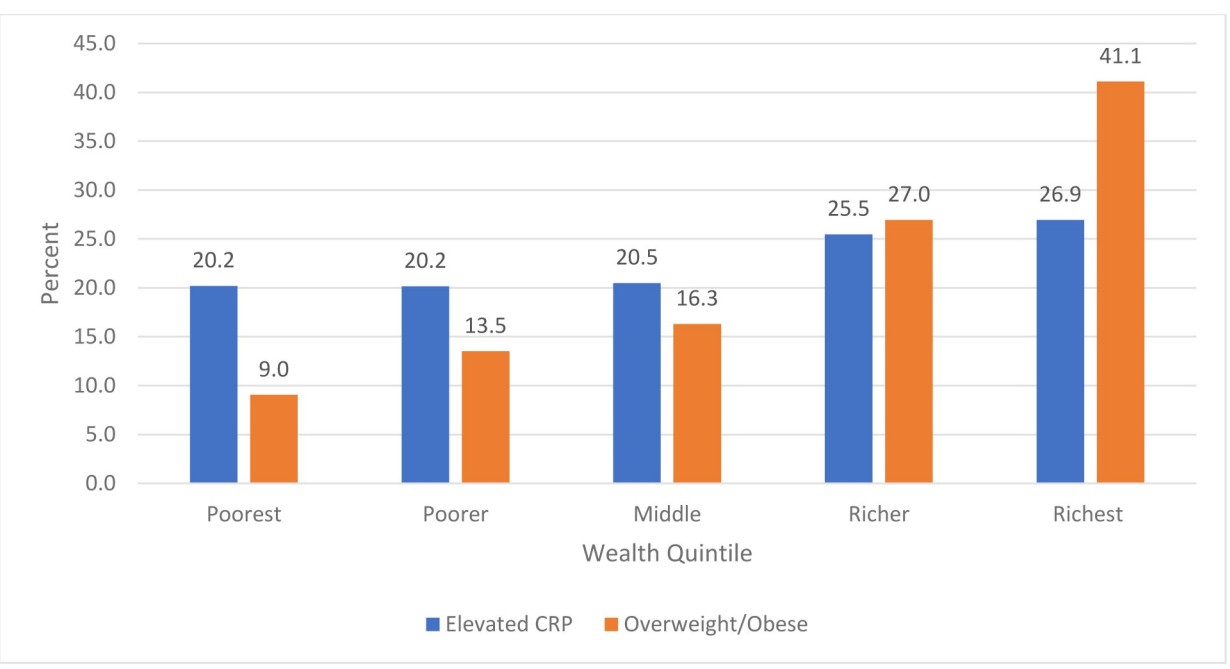

**Fig 3. Proportion of participants that have elevated CRP or are overweight/obese by wealth quintiles.**

administrative boundary to a novel indicator of urbanicity using a satellite-derived built environment index. Urban-rural designations each display some heterogeneity within and homogeneity between each other regarding the level of built environment, with much more variation among the rural defined areas. Urbanicity as measured by the binary administrative variable as well as the more granular built environment variable seems to be associated with increases in BMI and to a lesser extent, with CRP. BMI increased with increasing proximity to an urban center.

One of the objectives of this study was to examine the potential for using GHSL data to help explain markers of chronic disease by detecting levels of the built environment. To this end, we found that our built environment measure was associated with increases in BMI. In an adjusted model, as built environment categories increased BMI also increased. Additionally, there was an association between distance to a major urban center and BMI, with BMI decreasing as distance increased. In the urban vs rural category model, we found an association in the unadjusted but not the adjusted model. This suggests that using binary administrative categories of 'urban' or 'rural' does not provide enough detail to capture the relationship with BMI.

When explored separately by administrative urban or rural category, we found that built environment was associated with both CRP and BMI among rural areas but not urban ones. This may be because the urban defined areas are almost entirely in the most built up category and more saturated. We also found that distance to the nearest city was associated with BMI among rural areas. This suggests that more of the change is happening in areas that are developing from rural to urban, or rural areas that are building up. It will be critical to monitor changes over time and space and the potential for adverse health effects, as areas transform to more urban, and particularly if urban growth is not planned.

We also aimed to tease apart the joint effects of location and wealth. In models using both a binary urban/rural variable as well as the built environment categories, we found a strong effect of wealth on BMI. The measurement of wealth is so closely associated with urban

residence that it makes it difficult to disentangle the two from one another. As the population of Sub-Saharan African becomes more urban than rural, it will be important for surveys like the DHS to be able to use more refined measures of wealth with sensitivity to capture a full range of urban welfare including that cities are almost always more expensive to live in than their rural counterparts [31]. We found both elevated CRP and overweight/obesity decreased from the richest to poorest quintiles, but this decrease was more linear for weight status than for CRP.

In the present study, marriage and cohabitation was significantly associated with increased levels of both CRP and BMI, after controlling for other factors. Another study focusing on the Dar es Salaam region in Tanzania found that marriage/cohabitation was a risk factor for obesity among a cohort including women and men [41]. However, a study in Malawi found being married was weakly associated with having a lower risk of elevated CPR level [42]. One possibility for our finding is that marriage could be a proxy for having/raising children, which may be driving the association. Results have linked number of offspring to risk of cardiovascular disease, although the mechanisms are not clear [43]. Further analysis could explore this hypothesis, including whether the effect is associated with age and parity of mother as well. This would be important for understanding how within-household dynamics affect BMI and how these dynamics vary by urban or rural location.

In interpreting findings from this study, it is important to consider some of the limitations from using biomarker and locational data. One challenge is that we only have measurements from single time points for built environment (2014) and outcome measures of CRP and BMI have thus far only been collected during one DHS survey in 2010, prohibiting us from assessing their change over time as well as from making causal inferences. Further, we used a single measure of CRP to indicate chronic inflammation while repeated measures are understood to be better to rule out increased levels due to infection [44]. It could be useful to link biomarker data with health outcomes data such as hypertension, medication use, and self-rated health, as these would be helpful covariates and/or potential mediators between living in an urban environment and our health outcomes. This would also help us understand why there was a less clear association found between urbanicity and levels of CRP. Although we excluded pregnant women from our sample, it is possible that we were unable to distinguish elevated levels of CRP independent of other acute health conditions, as we did not have information on many relevant health conditions. Finally, intentional dislocation of the DHS data compromises the ability to construct environmental variables [45], in particular those at a fine resolution such as the GHSL, and within urban areas where large substantive changes can occur across very small geographic areas.

## Conclusions

Despite limitations, this study is the first known examination of urbanicity and chronic disease health indicators within Sub-Saharan Africa. First, our analysis provides prevalence of elevated levels of CRP and overweight/obesity among women in Tanzania. Second, we demonstrate that urbanicity is positively associated with BMI and that this association is partly but not fully accounted for by wealth. Finally, our study is among the first public health studies to use the sentinel corrected GHSL data, and as such is on the forefront of the use of satellite imagery to understand the effect of the built environment on health outcomes. These data help to define a continuum of urbanization, allowing health and social science researchers to no longer rely on limited data structures pertaining to urbanization that are seemingly inherent in the survey and administrative records of many countries. This study provides information about growing health concerns in an under-studied population and offers a unique methodological approach

to help understand how to think about and measure urbanicity in an increasingly urban world.

## Supporting information

**S1 Fig. Degree of urbanicity across wealth categories, by urban/rural classification.** (DOCX)

## Acknowledgments

The authors would like to acknowledge Cara Kraus-Perrotta for her support updating the literature review and reviewing the manuscript, and to thank Hasim Engin for his programming support.

The authors would also like to thank the CUNY Institute for Demographic Research for the Demography Fellowship awarded to author CM that facilitated early stages of this research. Lastly, thank you to the panelists and community feedback received when this initial work was presented at the 2016 Annual Meeting of Population Association of America.

## Author Contributions

**Conceptualization:** Carrie W. Mills, Deborah Balk.

**Formal analysis:** Jessie Pinchoff, Carrie W. Mills.

**Investigation:** Jessie Pinchoff.

**Methodology:** Jessie Pinchoff, Carrie W. Mills, Deborah Balk.

**Resources:** Deborah Balk.

**Software:** Jessie Pinchoff.

**Supervision:** Deborah Balk.

**Writing – original draft:** Jessie Pinchoff, Carrie W. Mills, Deborah Balk.

**Writing – review & editing:** Jessie Pinchoff, Carrie W. Mills, Deborah Balk.

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
