## [Decision Letter · Decision Letter 0]

11 Sep 2020

PONE-D-20-22046

Urbanization and health: the effects of the built environment on chronic disease risk factors among women in Tanzania

PLOS ONE

Dear Dr. Pinchoff,

Thank you for submitting your manuscript to PLOS ONE. After careful consideration, we feel that it has merit but does not fully meet PLOS ONE’s publication criteria as it currently stands. Therefore, we invite you to submit a revised version of the manuscript that addresses the points raised during the review process.

We look forward to receiving your revised manuscript.

Kind regards,

Sze Yan Liu, PhD

Academic Editor

PLOS ONE

2. Thank you for including your funding statement;

Reviewers' comments:

Reviewer's Responses to Questions

**Comments to the Author**

1. Is the manuscript technically sound, and do the data support the conclusions?

Reviewer #1: Yes

Reviewer #2: Yes

2. Has the statistical analysis been performed appropriately and rigorously? 

Reviewer #1: Yes

Reviewer #2: Yes

3. Have the authors made all data underlying the findings in their manuscript fully available?

Reviewer #1: Yes

Reviewer #2: Yes

4. Is the manuscript presented in an intelligible fashion and written in standard English?

Reviewer #1: Yes

Reviewer #2: Yes

5. Review Comments to the Author

Reviewer #1: The article under review explores the association between urbanicity and chronic disease risk factors or markers in Tanzania. The article contributes to the literature by analyzing these measures in an understudied geographic location, Tanzania. It also is innovative in its use of multiple measures of urbanicity, including the use of a nuanced satellite-based measure of the built environment. Results indicated that there were significant associations between these risk factors, particularly BMI, and urbanicity based on the satellite-derived measure. My comments are organized into 2 main categories, the first focusing on content and the second focusing on more minor editorial comments relating to organization and grammar.

Content:

Abstract lines 40-42: Please review the interpretation of your results carefully. The outcomes were log CRP and log BMI. So, I believe the increase of 0.43 and 0.024 between the lowest and highest built environment categories should be in the units of the outcome rather than percentage points.

Abstract line 33: This is more of a technicality, but BMI is more than just a measure of obesity (it measures weight in relation to height and is used to define a wide range of weight categories including underweight, normal weight, etc with high BMI indicating obesity). Perhaps space limitations make it difficult to explain fully in the abstract, but it would be useful to describe how BMI is defined and used to classify obesity in the text.

Methods: The choice of 2014 for the satellite data when the outcome data in the Tanzania DHS is 2010 means the exposure automatically follows the outcome. Did you also consider doing a sensitivity analysis with the earlier time point, 2000? Understandably 2000 is farther away (although it would still likely be correlated with built environment levels in 2010), and there will be continuity such that 2014 built environment may not be substantially different from 2010, But even if no sensitivity analysis was conducted, I think it is important to expand on the discussion beyond the fact that there isn’t longitudinal data for changes over time to more explicitly mention the issue with the temporal order between the exposure (2014) and outcome (2010).

Line 181: You mention that you did analyses using the more sensitive CRP cutoff. Were the results generally similar?

Line 236: The description of Table 2 seems to be a bit inaccurate. The table seems to present not just measures of satellite-derived urbanity but also other measures, i.e. DHS classification of urbanicity

The wording of the figure titles appears to be switched. For instance, “Figure 3. Wealth quintiles by proportion of participants that have elevated CRP or are overweight/obese” seems to make more sense as “Proportion of participants that have elevated CRP or are overweight/obese by wealth quintiles”

Similarly, the supplemental figure seems to be showing the distribution of the built environment (the boxplots along the y axis) by wealth quintile for urban/rural classification

Lines 354-356: Is there any literature that can be cited linking those factors to your health outcomes, particularly as precursors? It may be helpful to refer to some studies here (though I would think BMI may be on the pathway to hypertension, rather than hypertension being the mediator).

Editorial:

Introduction, line 54: The comma is not needed after the word “While”

Lines 80-81: Consider changing the wording since both “expected” and “expect” appear in the sentence. It may also be useful to explicitly say that they will experience “increases” in chronic disease outcomes since there is some level of these outcomes existing already.

Line 82: The verb “is” should be changed to “are” to describe scant descriptive statistics. You may also want to consider moving that sentence to right before the sentence “One very small study…” to improve the flow since they both focus on Tanzania while the sentences in between are about Africa more generally.

Line 88: The BMI acronym is introduced before spelling out what it stands for in line 90, so this should be reversed. (This section may also also be an appropriate place to mention what BMI is and how it is used to define obesity)

Line 120: the “s” can be removed from the word “allows”

Methods: This section could benefit from an additional subheading (i.e. Sample) at the paragraph beginning with line 140. The sentence on observations being weighted could also be moved to Statistical Analysis since that happens during the analysis phase.

Line 153: The word "been" can be removed from “previous work has been shown…” The sentence also kind of reiterates the discussion in the intro, so maybe a transition phrase will be helpful, such as “As discussed earlier,… This study therefore also explores a second measure…”

Line 158: It may be better to move the citation to the end of the complete sentence.

Line 221: There seems to be a missing decimal point in 230% with elevated CRP.

Line 245: Change “farther” to “farthest”. It would also be helpful to indicate in that sentence that CRP was not significant and wasn’t changing in a stepwise fashion.

Line 257: It may be useful to include “log” in front of CRP here. Also ensure that log is included when discussing BMI coefficients.

Line 267: The word “in” can be removed from “…as measured in by the DHS”.

Table 1: The units for BMI should be included: kg/m^2.

Line 282: The word “with” is missing from the phrase “…was not associated CRP”.

Tables 4A & B: The heading for ci95 should be defined (either as a note) or spelled out fully as 95% Confidence Interval. The headings are also justified differently in the two tables (centered vs. left)

Line 287-290: It may not be necessary to include the negative signs since you are saying that BMI decreased.

Line 296: Either “such as” or “particularly” could be removed from the parenthetical phrase “(such as particularly roof…)”.

Line 318: I would suggest being cautious using the word “predict” because the temporal order of the measures is the exposure is at a later date than the outcome. It would be better to say it is associated with increases.

Line 322: The “ly” can be removed from “administratively”. The word “rural” should also be added to the sentence.

Line 342: The word “that” can be removed before “found” in this sentence.

Line 367: In this sentence, are you referring to the interaction between urbanicity & wealth? The word “modified” usually suggests interaction or effect modification, which the results indicated was not significant. Are you instead saying that the association between urbanicity and BMI is partly explained by wealth (such that adding the wealth covariate to the model decreased the effect of urbanicity)?

Reviewer #2: The manuscript is well written and quite insightful. I have very minor corrections and clarifications that need to be addressed. Delete in following "...measured" Page 14, Line 267. On the same page. In Table 4A I think the first row, columns 2 and 4 the words "rural" and "urban" should each start with a capital letter. I think "ci95" in the same table has not been explained. I suppose it refers to 95% confidence interval which is conventionally presented as "95% CI". This comment applies to Table 4B as well.

6. PLOS authors have the option to publish the peer review history of their article (what does this mean?). If published, this will include your full peer review and any attached files.

Reviewer #1: No

Reviewer #2: **Yes: **Gobopamang Letamo

---

## [Author Response · Author response to Decision Letter 0]

29 Sep 2020

Content:

Abstract lines 40-42: Please review the interpretation of your results carefully. The outcomes were log CRP and log BMI. So, I believe the increase of 0.43 and 0.024 between the lowest and highest built environment categories should be in the units of the outcome rather than percentage points.

Yes thank you, we re-wrote to specify the correct units.

Abstract line 33: This is more of a technicality, but BMI is more than just a measure of obesity (it measures weight in relation to height and is used to define a wide range of weight categories including underweight, normal weight, etc with high BMI indicating obesity). Perhaps space limitations make it difficult to explain fully in the abstract, but it would be useful to describe how BMI is defined and used to classify obesity in the text.

Thank you for this comment, we have added language into the abstract and background clarifying BMI as a measure and describing how it relates to classification of overweight/obese for our study. In the methods section we have the definition as weight for height and that cut offs indicate different levels – with the highest cut off above 30 considered obese stated, but have clarified earlier in the paper. Throughout the paper we clarified when we were examining overweight and obesity, and when we were modeling BMI as a continuous variable.

Methods: The choice of 2014 for the satellite data when the outcome data in the Tanzania DHS is 2010 means the exposure automatically follows the outcome. Did you also consider doing a sensitivity analysis with the earlier time point, 2000? Understandably 2000 is farther away (although it would still likely be correlated with built environment levels in 2010), and there will be continuity such that 2014 built environment may not be substantially different from 2010, But even if no sensitivity analysis was conducted, I think it is important to expand on the discussion beyond the fact that there isn’t longitudinal data for changes over time to more explicitly mention the issue with the temporal order between the exposure (2014) and outcome (2010).

The 2014 GHSL is more accurate and is extremely similar to the 2000 GHSL values (mean difference of 1.9 percentage points, with over 65% of locations having no change) because 2014 was used to ‘back correct’ the 2000 image.

GHSL 2014 represents any built-up area by 2014, derived from Landsat scenes of any year before then. Similarly, 2000 represents built-up by 2000. Because our study country is tropical, it is most likely that the 2014 data set is a composite of images up to that point from many years. (If change were evenly distributed in all years, it would almost certainly happen by 2010. We don’t know when in the 2000-2014 interval the last observed change is observed to happen).

GHSL 2014 also uses higher resolution data (from sentinel satellite missions) to help refine the resolution, and in cloud-prone areas it provides more data in order to improve detection. As the sentinel satellites are new (2015+), the presence of built in 2015 is used as means for back casting the earlier product years with the machine learning algorithm and presence of other data layers. For this reason, the 2014 data product is most likely a superior data product than the earlier years, and thus while it post-dates our survey observation, we determine it to be the best proxy for built-settlement around this time. 

We have made changes to the text to summarize these points. 

Line 181: You mention that you did analyses using the more sensitive CRP cutoff. Were the results generally similar?

Yes the results are quite similar and we have added a line in the paper to note this.

Line 236: The description of Table 2 seems to be a bit inaccurate. The table seems to present not just measures of satellite-derived urbanity but also other measures, i.e. DHS classification of urbanicity

Thank you for catching this, we have revised to state the table presents findings by “three different measures of urbanicity” – one of these is satellite derived, but the other is DHS classification, and the third is the distance to an urban center.

The wording of the figure titles appears to be switched. For instance, “Figure 3. Wealth quintiles by proportion of participants that have elevated CRP or are overweight/obese” seems to make more sense as “Proportion of participants that have elevated CRP or are overweight/obese by wealth quintiles”

Agreed, we have updated the table title.

Similarly, the supplemental figure seems to be showing the distribution of the built environment (the boxplots along the y axis) by wealth quintile for urban/rural classification

We have revised the title accordingly.

Lines 354-356: Is there any literature that can be cited linking those factors to your health outcomes, particularly as precursors? It may be helpful to refer to some studies here (though I would think BMI may be on the pathway to hypertension, rather than hypertension being the mediator).

Thank you for this comment we have added a reference in this paragraph.

Editorial:

Introduction, line 54: The comma is not needed after the word “While”

The comma has been deleted.

Lines 80-81: Consider changing the wording since both “expected” and “expect” appear in the sentence. It may also be useful to explicitly say that they will experience “increases” in chronic disease outcomes since there is some level of these outcomes existing already.

The sentence has been revised. 

Line 82: The verb “is” should be changed to “are” to describe scant descriptive statistics. You may also want to consider moving that sentence to right before the sentence “One very small study…” to improve the flow since they both focus on Tanzania while the sentences in between are about Africa more generally.

We have edited to improve the flow in this paragraph.

Line 88: The BMI acronym is introduced before spelling out what it stands for in line 90, so this should be reversed. (This section may also also be an appropriate place to mention what BMI is and how it is used to define obesity)

We edited the sentence in line 87, and add more detail in line 90 (now line 91-92) to define obesity.

Line 120: the “s” can be removed from the word “allows”

Deleted.

Methods: This section could benefit from an additional subheading (i.e. Sample) at the paragraph beginning with line 140. The sentence on observations being weighted could also be moved to Statistical Analysis since that happens during the analysis phase.

Thank you, we have added a subheading (Data) and moved language related to weighting to the analysis section.

Line 153: The word "been" can be removed from “previous work has been shown…” The sentence also kind of reiterates the discussion in the intro, so maybe a transition phrase will be helpful, such as “As discussed earlier,… This study therefore also explores a second measure…”

We have edited this.

Line 158: It may be better to move the citation to the end of the complete sentence.

Agree- we have moved the citation to the end of the sentence.

Line 221: There seems to be a missing decimal point in 230% with elevated CRP.

Yes, thank you! Added.

Line 245: Change “farther” to “farthest”. It would also be helpful to indicate in that sentence that CRP was not significant and wasn’t changing in a stepwise fashion.

We changed the word farther to farthest, and added that CRP was not significant.

Line 257: It may be useful to include “log” in front of CRP here. Also ensure that log is included when discussing BMI coefficients.

We have added log where necessary.

Line 267: The word “in” can be removed from “…as measured in by the DHS”.

The word ‘in’ has been removed.

Table 1: The units for BMI should be included: kg/m^2.

We have added both units.

Line 282: The word “with” is missing from the phrase “…was not associated CRP”.

The word ‘with’ has been added.

Tables 4A & B: The heading for ci95 should be defined (either as a note) or spelled out fully as 95% Confidence Interval. The headings are also justified differently in the two tables (centered vs. left)

We have spelled out 95% confidence interval and centered both table headings to match.

Line 287-290: It may not be necessary to include the negative signs since you are saying that BMI decreased.

We have deleted the negative signs in this paragraph.

Line 296: Either “such as” or “particularly” could be removed from the parenthetical phrase “(such as particularly roof…)”.

We have edited this sentence.

Line 318: I would suggest being cautious using the word “predict” because the temporal order of the measures is the exposure is at a later date than the outcome. It would be better to say it is associated with increases.

Yes! We have edited this section.

Line 322: The “ly” can be removed from “administratively”. The word “rural” should also be added to the sentence.

We have made these two edits. 

Line 342: The word “that” can be removed before “found” in this sentence.

We have deleted the word “that”.

Line 367: In this sentence, are you referring to the interaction between urbanicity & wealth? The word “modified” usually suggests interaction or effect modification, which the results indicated was not significant. Are you instead saying that the association between urbanicity and BMI is partly explained by wealth (such that adding the wealth covariate to the model decreased the effect of urbanicity)?

Yes, we have edited accordingly. 

Reviewer #2: The manuscript is well written and quite insightful. I have very minor corrections and clarifications that need to be addressed. Delete in following "...measured" Page 14, Line 267. On the same page. In Table 4A I think the first row, columns 2 and 4 the words "rural" and "urban" should each start with a capital letter. I think "ci95" in the same table has not been explained. I suppose it refers to 95% confidence interval which is conventionally presented as "95% CI". This comment applies to Table 4B as well.

Thank you! We have edited page 14, line 267. For the tables, we have capitalized urban/rural and spelled out 95% confidence intervals for clarity.

---

## [Decision Letter · Decision Letter 1]

21 Oct 2020

Urbanization and health: the effects of the built environment on chronic disease risk factors among women in Tanzania

PONE-D-20-22046R1

Dear Dr. Pinchoff,

We’re pleased to inform you that your manuscript has been judged scientifically suitable for publication and will be formally accepted for publication once it meets all outstanding technical requirements. 

Please do note the editorial/grammatical edits noted by the reviewer.

Kind regards,

Sze Yan Liu, PhD

Academic Editor

PLOS ONE

Additional Editor Comments (optional):

Reviewers' comments:

Reviewer's Responses to Questions

**Comments to the Author**

1. If the authors have adequately addressed your comments raised in a previous round of review and you feel that this manuscript is now acceptable for publication, you may indicate that here to bypass the “Comments to the Author” section, enter your conflict of interest statement in the “Confidential to Editor” section, and submit your "Accept" recommendation.

Reviewer #1: (No Response)

2. Is the manuscript technically sound, and do the data support the conclusions?

Reviewer #1: Yes

3. Has the statistical analysis been performed appropriately and rigorously? 

Reviewer #1: Yes

4. Have the authors made all data underlying the findings in their manuscript fully available?

Reviewer #1: Yes

5. Is the manuscript presented in an intelligible fashion and written in standard English?

Reviewer #1: Yes

6. Review Comments to the Author

Reviewer #1: I recommend this paper for publication but have noted some additional minor changes that should still be made during the proofing stage. All line numbers reference the clean version of the revised manuscript:

In some of the places where edits were made, the clean version seems to have a wide space (i.e. introduction “…and thus it is expected that these regions will experience increases in chronic health including diabetes and cardiovascular disease”). A similar thing happens in lines 275, 278, 324, and 340 of the clean version, so it would be good to check for any extra spaces that are left in the proof stage.

In lines 118-120 of the clean version, the “s” should be added back to “allow” to make it allows in the statement: “Similarly, recent work by the European Commission finds that by combining satellite and census data, a degree of urbanization metric allow for understanding the urban continuum”. My previous comment only applied to the word allows in the following sentence where the s needed to be removed, and that has been corrected.

Although your response indicates a change, Table 1 still doesn’t seem to include the units for BMI, so that would be helpful.

In line 263 of the clean version add “log” to the statement “…those in the highest built up category had CRP increased by 0.43” before CRP.

Line 302, make “E.g.” all lower case (e.g.

Line 349, the word “that” was not removed in the revision as indicated but is not necessary: “However, a study in Malawi that found being married was weakly associated with having a lower risk of elevated CPR level.”

The negative signs can also be removed from lines 279-280 since you are already talking about a decrease: “…log BMI decreased by -0.02 kg/m2 in the unadjusted model (p<0.05) and decreased by -0.02 kg/m2 (p<0.1) in the adjusted model”

Also, I would also consider removing the kg/m2 in the sentence since I am not sure that the units are the same when the outcome is in log form. I think a safer word choice would be “units”, “log units”, or using nothing at all. The same goes for the abstract where you say “log CRP increased 0.43 μg/mL in the highest built up areas compared to not built up (p<0.05); log BMI increased 0.02 kg/m2”. I might refrain from using units here because of the transformation (or substitute the word “units” instead). I looked at several resources about interpreting log-transformed models (see below), and while I don’t think you need to convert to percent and change the sentences, if you are using the coefficients as is, it might be safter not to assume that the units of measure are the same since they arenow log units:

https://data.library.virginia.edu/interpreting-log-transformations-in-a-linear-model/

https://stats.idre.ucla.edu/other/mult-pkg/faq/general/faqhow-do-i-interpret-a-regression-model-when-some-variables-are-log-transformed/

https://www.stata.com/stata-news/news34-2/spotlight/

7. PLOS authors have the option to publish the peer review history of their article (what does this mean?). If published, this will include your full peer review and any attached files.

Reviewer #1: No

---

## [Editor Report · Acceptance letter]

23 Oct 2020

PONE-D-20-22046R1 

Urbanization and health: the effects of the built environment on chronic disease risk factors among women in Tanzania 

Dear Dr. Pinchoff:

I'm pleased to inform you that your manuscript has been deemed suitable for publication in PLOS ONE. Congratulations! Your manuscript is now with our production department. 

Kind regards, 

on behalf of

Dr. Sze Yan Liu 

Academic Editor

PLOS ONE